# Ciliary Neurotrophic Factor Modulates Multiple Downstream Signaling Pathways in Prostate Cancer Inhibiting Cell Invasiveness

**DOI:** 10.3390/cancers14235917

**Published:** 2022-11-30

**Authors:** Giovanni Tossetta, Sonia Fantone, Rosaria Gesuita, Gaia Goteri, Martina Senzacqua, Fabio Marcheggiani, Luca Tiano, Daniela Marzioni, Roberta Mazzucchelli

**Affiliations:** 1Department of Experimental and Clinical Medicine, Università Politecnica delle Marche, 60126 Ancona, Italy; 2Centre of Epidemiology, Biostatistics and Medical Information Technology, Università Politecnica delle Marche, 60126 Ancona, Italy; 3Department of Biomedical Sciences and Public Health, Università Politecnica delle Marche, 60126 Ancona, Italy; 4Department of Biomedical Sciences and Public Health, Section of Pathological Anatomy, Università Politecnica delle Marche, 60126 Ancona, Italy; 5Department of Life and Environmental Sciences, Università Politecnica delle Marche, 60131 Ancona, Italy

**Keywords:** CNTF, CNTFRα, prostate cancer, LNCaP, 22Rv1, castration resistant, castration sensitive, STAT3, ERK, MMP

## Abstract

**Simple Summary:**

Prostate cancer (PCa) is one of the major cancers affecting men. Localized and loco-regional PCa is usually treated by radical prostatectomy, radiation therapy, cryosurgery or with HIFU (High-Intensity Focused Ultrasound). Advanced or metastatic cancers are most often treated with androgen deprivation therapy, but too often such patients progress to lethal castration-resistant PCa. IL-6 plays a key role in prostate cancer but no data on ciliary neurotrophic factor (CNTF), a member of interleukin-6 cytokine family, are known. We detected CNTF and its receptor CNTFRα in human androgen-responsive and in castration-resistant prostate cancer (CRPC) tissues. In addition, we showed that CNTF downregulated MMP-2 and GLUT-1 expression by MAPK/ERK, AKT/PI3K and Jak/STAT pathways in a CRPC in vitro model. This suggests a pivotal role of CNTF as negative modulator of invasion processes in this PCa model.

**Abstract:**

Background: Prostate cancer (PCa) remains the most common diagnosed tumor and is the second-leading cause of cancer-related death in men. If the cancer is organ-confined it can be treated by various ablative therapies such as RP (radical prostatectomy), RT (radiation therapy), brachytherapy, cryosurgery or HIFU (High-Intensity Focused Ultrasound). However, advanced or metastatic PCa treatment requires systemic therapy involving androgen deprivation, but such patients typically progress to refractory disease designated as castration-resistant prostate cancer (CRPC). Interleukin-6 (IL-6) has been established as a driver of prostate carcinogenesis and tumor progression while less is known about the role of ciliary neurotrophic factor (CNTF), a member of the IL-6 cytokine family in prostate cancer. Moreover, MAPK/ERK, AKT/PI3K and Jak/STAT pathways that regulate proliferative, invasive and glucose-uptake processes in cancer progression are triggered by CNTF. Methods: We investigate CNTF and its receptor CNTFRα expressions in human androgen-responsive and castration-resistant prostate cancer (CRPC) by immunohistochemistry. Moreover, we investigated the role of CNTF in proliferative, invasive processes as well as glucose uptake using two cell models mimicking the PCa (LNCaP cell line) and CRPC (22Rv1 cell line). Conclusions: Our results showed that CNTF and CNTFRa were expressed in PCa and CRPC tissues and that CNTF has a pivotal role in prostate cancer environment remodeling and as a negative modulator of invasion processes of CRPC cell models.

## 1. Introduction

Prostate cancer is one of the major cancers affecting men and remains the second leading cause of cancer-related death [1]. It is a biologically heterogeneous disease; the majority of instances are localized or loco-regional and can be treated by surgery or other ablative procedures, while more advanced or metastatic patients who remain androgen-dependent at the time of diagnosis are treated with androgen deprivation therapy (ADT). The response to ADT is variable and many such patients become refractory to treatment and will progress to lethal castration-resistant prostate cancer (CRPC) [2]. IL-6 has been established as a driver of carcinogenesis and tumor progression in PCa [3,4,5], while less is known about the role of ciliary neurotrophic factor (CNTF), a member of IL-6 type cytokine family [6]. CNTF, and other cytokines of this group (e.g., IL-6, LIF, OMS and IL-11), require additional non-signaling receptors (long signal-transducing α-receptor glycoprotein chains) that are bound to the membrane [7]. CNTFRα-chain non-signaling receptor subunit (CNTFRα) is the high affinity non-signaling receptor for CNTF. CNTFRα is associated with a heterodimeric β-receptor complex LIFR and gp130, and this association allows signal transduction due to phosphorylation of tyrosine residue of CNTFRα after its interaction with CNTF [7]. The CNTF receptor complex triggers simultaneous activation of different intracellular signaling pathways such as MAPK/ERK, AKT/PI3K and Jak/STAT pathways which mediate survival and/or differentiation in different cell types [8,9]. Moreover, it has been reported that PCa progression and metastasis are correlated both to PTEN/AKT/PI3K axis alteration and RAS/MAPK/ERK signaling activation, while the RAS/MAPK/ERK pathway alone is significantly activated in both primary and metastatic lesions [10,11]. In addition, similar to many other tumors, glucose metabolism in PCa is different between the early and advanced stages of the disease. Moreover, the glycolytic metabolic profile differs in androgen-sensitive and insensitive PCa cells [12,13]. There are fourteen different members of GLUT transporters (GLUT 1–12, GLUT14, and H/myo-inositol transporter) and different pattern of protein expression or membrane translocation are observed in relation to tumor-specific glycolytic pathway modulation. Therefore, an increase in glucose uptake has been associated mainly with GLUT1 overexpression, but may also involve other GLUTs, including GLUT4 [14,15]. It has been demonstrated that an elevated AKT activity is involved in the high rate of glucose uptake in cancer cells by GLUT-1 [16,17,18] and that treatment with an AKT-specific inhibitor caused degradation of GLUT-1 in sustained AKT-activated breast cancer cells [19]. Human studies suggested that high levels of GLUT-1 expression in tumors are associated with poor survival [20,21]. The presence of CNTF and CNTFRα in the basal cell layer (in which reside stem cells) in the prostatic normal glandular epithelium has been previously demonstrated in our previous investigation [6]. Thus, the aim of this study is to detect the localization of CNTF and CNTFRα in androgen-sensitive and androgen-insensitive human PCa tissues and to analyze the possible role of CNTF in signaling-pathway modulation involved in glucose uptake, proliferation, migration and invasion processes using sensitive (LNCaP) and androgen-insensitive (22Rv1) cell lines.

## 2. Materials and Methods

### 2.1. Tissue Collection

In this study, we analyzed a total of 20 human prostate carcinoma samples: 10 untreated PCa were obtained from radical prostatectomy and 10 CRPC were obtained from transurethral resection of the prostate (TURP). A pathologist (R.M.) reviewed and selected the hematoxylin–eosin stained samples that were used in this study. All the samples were collected from the Pathology Services of the Polytechnic University of the Marche Region United Hospitals. The procedures followed for the collection of samples were in accordance with the Helsinki Declaration of 1975, as revised in 2013. The permission of the Human Investigation Committee of Marche Region (IT) was granted (protocol number 2020/395).

### 2.2. Immunohistochemistry

All prostate samples (PCa and CRPC) were fixed in 10% neutral buffered formalin and routinely processed for paraffin embedding. Immunohistochemical staining was per-formed as previously described [22,23]. Briefly, after dewaxing, paraffin sections were rinsed in phosphate-buffered saline (PBS), incubated with 3% hydrogen peroxide for 40 min to block endogenous peroxidase. Pre-treatment by heat in 10 mM citrate buffer, pH 6.0 for 5 min was used for CNTF while pre-treatment by 100 ng/mL Proteinase K (Sigma-Aldrich, St. Louis, MO, USA) for 5 min at 37 °C was used for CNTFRα. After pre-treatment, sections were rinsed with PBS and incubated with normal horse serum (Vector laboratories, Burlingame, CA, USA) diluted 1:75 in PBS for 1 h at room temperature (RT). Sections were then incubated overnight with the primary antibodies (listed in Table 1) diluted in PBS at 4 °C. After a thorough rinse in PBS, sections were incubated with the appropriate biotinylated secondary antibody (Vector laboratories, Burlingame, CA, USA) diluted 1:200 *v*/*v* solution for 30 min at room temperature. Vectastain ABC Kit (Vector Laboratories, Burlingame, CA, USA) for 1h at room temperature and 3′,3′- diaminobenzidine hydrochloride (Sigma-Aldrich, St. Louis, MO, USA) were used to develop the immunohistochemistry reaction. Sections were counterstained with Mayer’s hematoxylin, dehydrated and mounted using Eukitt^®^ solution (Kindler GmbH and Co., Freiburg, Germany). Negative controls were performed by omitting the first or secondary antibody for all the immunohistochemical reactions performed in this study as procedure control. Negative controls were performed by replacing the primary or the secondary antibody with an isotype antibody: monoclonal rabbit IgG (ab172730; Abcam, Cambridge, UK) for CNTF and monoclonal mouse IgG2a (ab18415; Abcam) for CNTFRα as control specificity antibody.

### 2.3. Cell Culture

Androgen-dependent (LNCaP) and androgen-independent (22Rv1) cell lines (ATCC/LGC Standards, Manassas, VA, USA) were grown in RPMI 1640 (Life technologies, Carlsbad, CA, USA) supplemented with 10% Fetal Bovine Serum (FBS; Gibco, Thermo Fisher Scientific, Waltham, MA, USA) and 100 U/mL penicillin and streptomycin (Gibco) at 37 °C, 95% humidity and 5% CO_2_. The medium was changed every 2 to 3 days and cells were split 1:4 every 4 days.

### 2.4. Immunofluorescence

Immunofluorescence staining was performed as previously described [24,25]. Briefly, LNCaP and 22Rv1 cells were washed in Dulbecco’s PBS (Life technology, Monza, Italy), fixed in 4% paraformaldehyde in PBS for 10 min at room temperature (RT), and permeabilized in PBS 0.1 M added with 0.1% Triton X-100 (Sigma, Milano, Italy) for 5 min. After washing in PBS at RT, cells were blocked with 10% Normal Donkey Serum (Jackson ImmunoResearch, Pennsylvania, PA, USA) in PBS 0.1 M and incubated overnight at 4 ℃ with the primary antibodies listed in Table 1. Cells were then washed 3 times in PBS and incubated with the FITC-conjugated donkey anti-rabbit (for CNTF) and TRITC-conjugated anti-mouse (for CNTFRα) IgG secondary antibodies (both from Jackson ImmunoResearch, West Grove, PA, USA) 1:400 for 30 min at room temperature. TOTO3 (1:5000) probe was used for nuclear staining. Finally, the slides were cover-slipped with propyl gallate and evaluated with a Leica TCS-SL spectral confocal microscope. For negative controls, see the immunohistochemistry Section 2.2.

### 2.5. Western Blotting

Once LNCaP and 22Rv1 cells reached 80% confluence, cells were lysed by using the following lysis buffer: 0.1 M PBS, 0.1% (*w*/*v*) SDS, 1% (*w*/*w*) NONIDET-P40, 1 mM (*w*/*v*) Na orthovanadate, 1 mM (*w*/*w*) PMSF (phenyl methane sulfonyl fluoride), 12 mM (*w*/*v*) Na deoxycholate, 1.7 μg/mL Aprotinin, pH 7.5. Cell lysates were centrifuged at 20,000× *g* for 20 min at 4 °C and the supernatants were aliquoted and stored at −80 °C. Viable counts using the trypan blue dye exclusion test were routinely performed. All experiments were performed in duplicate and were repeated at least 3 times. The proteins’ concentrations were determined by a Bradford protein assay (Bio-Rad Laboratories, Milan, Italy). Protein samples were fractionated on 10% SDS-polyacrylamide gels (SDS-PAGE) and electrophoretically transferred (Trans-Blot^®^ Turbo™ Transfer System; Bio-Rad Laboratories Inc, Richmond, CA, USA) to nitrocellulose membranes. Non-specific protein binding was blocked with 5% (*w*/*v*) non-fat-dried milk (Bio-Rad Laboratories) in Tris-buffered saline (TBS/0.05% Tween 20 (TBS-T)) for 1h. Blots were incubated with the primary antibodies listed in Table 1 overnight at 4 °C. After washing, blots were incubated with anti-rabbit secondary antibody conjugated with horseradish peroxidase (Amersham Italia s.r.l., Milano, Italy) diluted 1:5000 in TBS-T. Detection of bound antibodies was performed with the Clarity Western ECL Substrate (Bio-Rad Laboratories) and images were acquired with Chemidoc (Bio-Rad Laboratories). Bands were analyzed using the ImageJ software (https://imagej.nih.gov/ij/download.html, accessed on 12 September 2022) for quantification, and normalization was completed using β-actin band intensities. Data represent the mean ± SEM, and were analyzed for statistical significance (*p* < 0.05) using the Student’s *t*-test. All experiments were performed in triplicate and were repeated at least three times.

### 2.6. CNTF Treatment of LNCaP and 22Rv1 Cell Lines

After verifying the presence of the CNTFRα, a dose/responsive curve was performed to test the best CNTF concentration showing a significant response for cellular treatments of LNCaP and 22Rv1 cell lines. These cell lines were grown in RPMI 1640 (Life technologies, Waltham, MA, USA) supplemented with 10% Fetal Bovine Serum (FBS; Gibco, Thermo Fisher Scientific, Waltham, MA, USA). They were not serum-starved, counted and plated in six well plates. These cells were treated with 0, 2, 10 and 20 ng/mL by recombinant human CNTF (rhCNTF) for 15 min, to detect which of the following signaling pathways was trigged: pERK/ERK; pAKT/AKT; and pSTAT3/STAT3. These signaling pathways and the expression of MMP-2, MMP-9, GLUT-1 and -4 were analyzed by Western blotting as described above, using the primary antibodies shown in Table 1. pSTAT3, pAKT and pERK1/2 quantities were normalized using total STAT3, AKT and ERK1/2, respectively, while GLUT-1, GLUT-4, MMP-2, MMP-9 and PCNA quantitates were normalized by β-actin. Results were calculated in arbitrary units (AU).

### 2.7. Analysis of Glucose Uptake after CNTF Treatment

The 22Rv1 cells were seeded in 96-well black plates (2 × 10^4^ cells per well), then the cells, untreated and treated with 10 ng/mL rhCNTF, were incubated at 37 °C for 24 h at 5% CO_2_. The cells were then incubated for 45 min with 50 µM 2-nitrobenzodeoxyglucose (2-NBDG) in glucose-free Dulbecco Modified Eagle’s Medium (DMEM) (Sigma, Milano, Italy). The level of cellular fluorescent 2-NBDG was evaluated at 550/590 nm with Synergy Ht fluorescence microplate reader (Biotek Instruments, VT, USA). Fluorescence values have been normalized for the number of live cells (by MTT analysis) and reported as Mean Fluorescence Intensity (MFI).

### 2.8. Wound Healing Assay 

The 22Rv1 cells showed MMP-2 downregulation if treated with rhCNTF. To investigate the role of CNTF in cell motility, these cells were grown to confluence and then scratch wounded with a sterile plastic micropipette tip. Then, the cells were rinsed 3 times with warm media to wash away scraped-off cells in the wound. The cells were untreated or treated with 10 ng/mL of rhCNTF in RPMI 1640 (Life technologies, Waltham, MA, USA) supplemented with 10% Fetal Bovine Serum (FBS; Gibco, Thermo Fisher Scientific, Waltham, MA, USA). Digital images of these cells were taken at 0, 4, 8, and 24 h. The area (Relative Wound Area) of the wound not occupied by cells was measured using a morphological imaging system ImageJ software (National Institutes of Health, http://imagej.nih.gov/ij accessed on 12 September 2022). The experiment was repeated at least 3 times.

### 2.9. Transwell Invasion Assay

Matrigel invasion assays were performed as previously reported [26]. Briefly, 24-well transwell inserts with 8.0 μm pores (#353097; Corning, NY, USA) were coated with 100 μL of 250 μg/mL LDEV-free Matrigel (#356234; Corning, NY, USA) diluted in serum-free RPMI 1640 medium according to the manufacturer’s instructions. The 22Rv1 cells were pretreated with 10 ng/mL CNTF for 24 h before being seeded in the transwell chambers. In the upper chambers, 5 × 10^4^ cells were seeded on top of the Matrigel and supplemented with 200 μL serum-free RPMI 1640, with/without 10 ng/mL CNTF. Then, 10% FBS in 750 μL of the medium was used as a chemoattractant in lower chambers. After 24 h, non-invading cells were gently removed from the apical side of the membrane by scrubbing with a cotton swab, whereas the cells remaining on the lower surface were fixed with 100% methanol for 20 min and stained with 0.5% crystal violet for 15 min. Cells in four non-overlapping fields were counted using ImageJ software. Data were expressed as mean cell number in each of the four fields. All experiments were repeated 3 times in triplicate.

### 2.10. Statistical Analysis 

The variables were not normally distributed in the Shapiro–Wilk test, and a non-parametric approach was used in the analysis. The distribution of fluorescent intensity (glucose uptake assay) and of invaded cells in control and CNTF groups were compared using the Wilcoxon rank-sum test. The nonparametric analysis of longitudinal data in factorial experiments [27] was used to compare the distribution of the relative wound area (dependent variable) between the control and CNTF groups (wound healing assay), to evaluate the distribution of the dependent variable over the four time points, and to analyze the interaction between the groups and the time points. Statistical analyses were carried out using R 4.0.5 (R Foundation for Statistical Computing; Vienna, Austria). The data are expressed as median and IQR. The significance level was set at 0.05.

## 3. Results

### 3.1. Human PCa and CRPC Tissues Expressed CNTF and CNTFRα

Neoplastic cells of PCa (Figure 1a,c) and CRPC (Figure 1b,d) samples were highly stained for CNTF (Figure 1a,b) and CNTFRα (Figure 1c,d). Stromal components (*) were weakly positive for CNTF and mainly negative for CNTFRα. Endothelial cells (arrow in d) in the majority of prostate vessels were positive or weakly stained for CNTFRα and CNTF, respectively, in all prostate samples analyzed.

### 3.2. LNCaP and 22Rv1 Cell Lines Expressed Both CNTF and CNTFRα 

Immunopositivity for CNTF (Figure 2, green staining) was both nuclear (on chromatin; the nucleoli are negative) and cytoplasmatic in LNCaP and 22Rv1 cell lines (Figure 2b,h). The red signal for CNTFRα was present in the cytoplasm and cell membranes with a high intensity in the perinuclear region (Figure 2e,k) in both cell lines. Channels merging is shown in Figure 2c,f,i,l.

### 3.3. MAPK/ERK, AKT/PI3K and Jak/STAT Pathways Modulation after CNTF Treatments

rhCNTF at 2 ng/mL induced pSTAT3 phosphorylation, while at 10 ng/mL it induced pAKT and pERK de-phosphorylation in both LNCaP and 22Rv1 cell lines (Figure 3a,b, respectively). The histograms representing the quantitative analysis of pSTAT3, pAKT and pERK expression levels normalized by STAT3, AKT and ERK, respectively, were depicted on the right side of the representative Western blotting. The statistical analysis showed a significant pSTAT3 increase after rhCNTF treatments in both cell lines (Figure 3a,b). No significant decrease in pAKT and pERK after treatments with 2 ng/mL of rhCNTF were shown in both cell lines (Figure 3a,b) while treatments with 10 ng/mL of rhCNTF showed a significant decrease in pAKT (*p* = 0.024; *p* = 0.0008) and pERK (*p* = 0.0007; *p* = 0.0009) in theLNCaP and 22Rv1 cell lines, respectively. The effect of rhCNTF treatments on GLUT-1, GLUT-4, MMP-2, MMP-9 and PCNA regulation is depicted in Figure 4. The histograms of the quantitative analysis of these molecules normalized by β-actin were shown at the right side of the representative Western blotting. No significant difference was shown for all the molecules analyzed in the LNCaP cell line (Figure 4a) and for MMP-9, GLUT-4 and PCNA in the 22Rv1 cell line (Figure 4b). On the contrary, rhCNTF treatments (10 ng/mL) induced a significant decrease in GLUT-1 (*p* = 0.0008) and MMP-2 (*p* = 0.03) in the 22Rv1 cell line (Figure 4b).

### 3.4. CNTF Treatments Did Not Affect Cell Migration and Glucose Uptake but Reduce 22Rv1 Cells Invasiveness

The LNCaP cell line was not considered for the following experiments because rhCNTF treatments did not significantly modify GLUT-1, GLUT-4, MMP-2, MMP-9 and PCNA expressions. In order to investigate the role of CNTF in cell motility, the 22Rv1 cell lines were treated with 10 ng/mL hrCNTF for 0, 4, 8 or 24 h. As reported in Figure 5, no significant differences in cell migration were found at any times between the untreated and hrCNTF-treated 22Rv1 cells suggesting that CNTF is not involved in modulating cell migration as represented in the box plots below the wound healing photos. 

Moreover, in order to evaluate a possible role of CNTF in modulating glucose uptake, we treated the 22Rv1 cell lines with 10 ng/mL hrCNTF for 24 h and performed a glucose uptake assay. As reported in Figure 6, no changes in glucose uptake were found between CNTF-treated and -untreated cells proving that CNTF does not alter glucose uptake in the 22Rv1 cell line. 

To evaluate the involvement of CNTF in 22Rv1 in invasiveness, we treated 22Rv1 cells with 10 ng/mL hrCNTF for 24 h. Interestingly, the distribution of the number of invaded cells is significantly (*p* < 0.001) lower in the cells treated with 10 ng/mL CNTF (median 89; IQR 71-98) compared to untreated cells (median 208; IQR 149-247) (Figure 7). 

## 4. Discussion

Despite many studies that have investigated CNTF and CNTFRα in various tissues and organs [26,28,29,30,31], there are no available data on the localization of these molecules and what role they have in PCa. 

In the present study, we investigated these issues using both androgen-dependent (LNCaP) and androgen-independent (22Rv1) cell lines to mimic untreated PCa and CRPC, respectively. CNTF and CNTFRα were expressed in untreated PCa and CRPC, suggesting that CNTF can have a paracrine-constitutive role as previously suggested for normal prostate tissue [6]. In addition, our in vitro data showed that exogenous CNTF administration can modulate JAK/STAT3, MAPK/ERK and PI3K/AKT signaling pathways in both PCa cell models. In particular, CNTF upregulated pSTAT3 but downregulated pAKT and pERK in both cell culture models. 

It is known that the activated ERK signaling pathway can facilitate the transformation of normal cells to malignant tumors [32] and that ERK1/2 activation is characteristic of CRPC [33]. In addition, Nickols and colleagues showed that frequent phosphorylation of ERK1/2 has been detected in metastatic CRPC [33]. In our hands, no modification was detected regarding PCNA expression, a proliferative marker. Therefore, we hypothesized that the upregulation of pSTAT3 could trigger apoptosis rather than a proliferative pathway by acting on Bcl-2/Bax balance as recently suggested [34,35]. In addition, we can speculate that the downregulation of pAKT and pERK due to CNTF can prevent cell proliferation and invasiveness, as our data demonstrated, masking the effect of pSTAT3 upregulation. 

It is known that cell survival is also ensured by glucose supply and, particularly rapid glucose transport in cancer cells facilitated by GLUT proteins spanning across the cell membrane. Although there are several types of GLUT proteins, the GLUT-1 and GLUT-4 proteins have been pinpointed to be those involved in glucose transport through the cell membranes of neoplastic cells [14,36]. In addition, androgens are intimately associated with the development of PCa [37] by acting as important metabolic regulators stimulating the glycolytic flux in neoplastic prostate cells [12,38,39,40]. Both androgen-responsive and androgen non-responsive PCa cells display a distinct glycolytic profile [12]. In agreement, our experiments showed that GLUT-1 was decreased in 22Rv1 cells (androgen-non-responsive) after CNTF treatments by inhibiting MAPK/ERK and AKT/PI3K pathways while no significant modulation of GLUT-1 was present in LNCaP cells (androgen-responsive). Recently, it has been demonstrated that overexpression of GLUT-1 not only accelerates glucose metabolism but also protects cancer cells from glucose deprivation-induced oxidative stress [41]. Other studies involving human subjects suggested that high levels of GLUT-1 expression in tumors were associated with poor survival [20,21] and that knockdown of GLUT-1 inhibited cell glycolysis and proliferation and led to cell cycle arrest at G2/M phase in the 22RV1 cell line [42]. Moreover, these authors demonstrated that GLUT-1 knockdown inhibits the tumor growth in the 22Rv1 cell line and that GLUT-1 knockdown has no effect on cell migration in vitro. Interestingly, our data demonstrated that CNTF treatments, although decreasing GLUT-1 expression, did not affect glucose uptake and neoplastic cell motility, confirming in part the previous data [42]. 

Our data showing the downregulation of MAPK/ERK and AKT/PI3K pathways by CNTF treatments suggest that this molecule may be involved in MMP modulation that is downstream of these activation pathways. Generally, proteolytic degradation of extra-cellular matrix (ECM) is mediated by MMPs, with MMP-2 and MMP-9 playing a critical role in prostate cancer progression [43,44,45]. It was demonstrated that MMP-2 and MMP-9 inhibition suppressed the metastatic potential of PCa cells [46]. Moreover, MMP-2 overexpression by malignant prostatic epithelia was associated with a poor disease-free survival [47,48]. Interestingly, our data showed that CNTF treatments significantly inhibited MMP-2 expression and our invasion assay demonstrated a significant inhibition of cell invasiveness in the 22Rv1 cell model mimicking CRPC. A limit of this study is the definitive proof that MMP-2 downregulation is responsible for significant inhibition of 22Rv1 cell invasiveness. Therefore, our findings should be corroborated by MMP-2 overexpression and silencing assays in 22Rv1 cells treated with CNTF.

## 5. Conclusions

We affirm that CNTF has a pivotal role in the prostate-cancer environment remodeling and suggest that this cytokine may be viewed as a modulator of invasion processes. In addition, we demonstrated that CNTF downregulated MMP-2 expression which is the main MMP involved in cell cancer invasiveness (Figure 8).

More studies are clearly required to corroborate our data, as in vivo male murine studies concerning allographic transplantation of 22Rv1 cells in CNTF-treated and -untreated conditions to evaluate the CNTF effect in tumor growth and metastasis inhibition. These further studies could lead to a novel therapeutic approach in patients with CRPC.

## Figures and Tables

**Figure 1 cancers-14-05917-f001:**
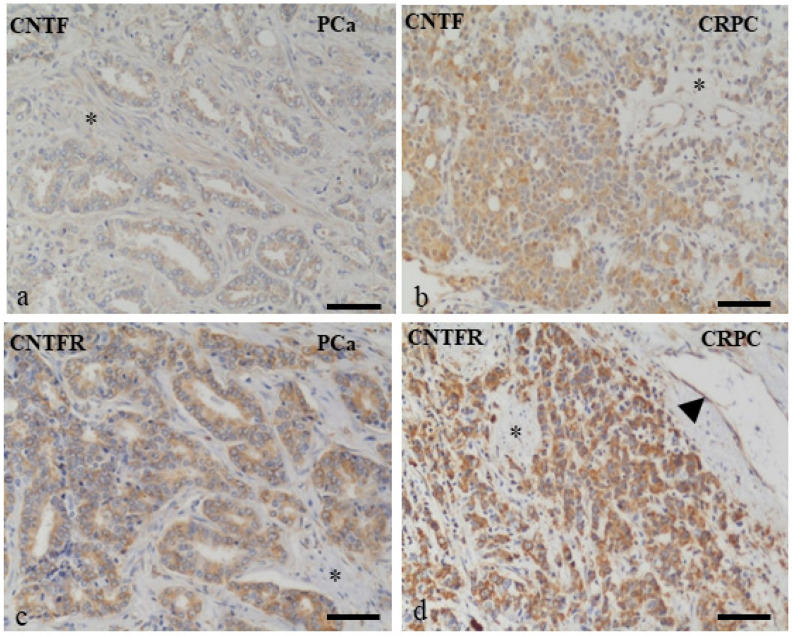
Immunohistochemistry localization of CNTF and CNTFRα in prostate cancer samples. CNTF (**a**,**b**) and CNTFRα (**c**,**d**) were expressed in PCa (**a**,**c**) and CRPC (**b**,**d**) tissues. Stromal components (*) were weakly positive for CNTF and mainly negative for CNTFRα. Endothelial cells (arrowhead in d) were positive or weakly stained for CNTFRα and CNTF, respectively. Scale bars: a,c = 100 μm; d,b = 200 μm.

**Figure 2 cancers-14-05917-f002:**
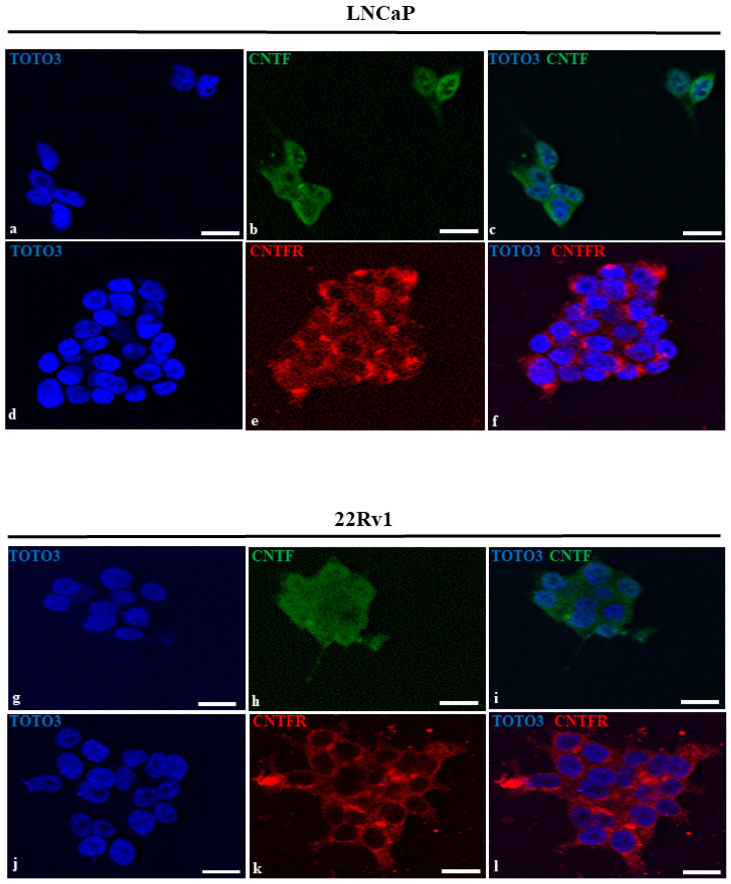
Immunofluorescence of CNTF and CNTFRα in LNCaP e 22Rv1cell lines. Nuclei (**a**,**d**,**g**,**j**) are stained in blue. CNTF (**b**,**h**) was stained in green and was mainly expressed in cytoplasm of both LNCaP and 22Rv1 cell lines. CNTFRα (**e**,**k**) was stained in red and was expressed in the cytoplasm and cell membranes with a high intensity in the perinuclear region of both cell lines. (**c**,**f**,**i**,**l**) represent merged channels. Scale bars: 30 μm.

**Figure 3 cancers-14-05917-f003:**
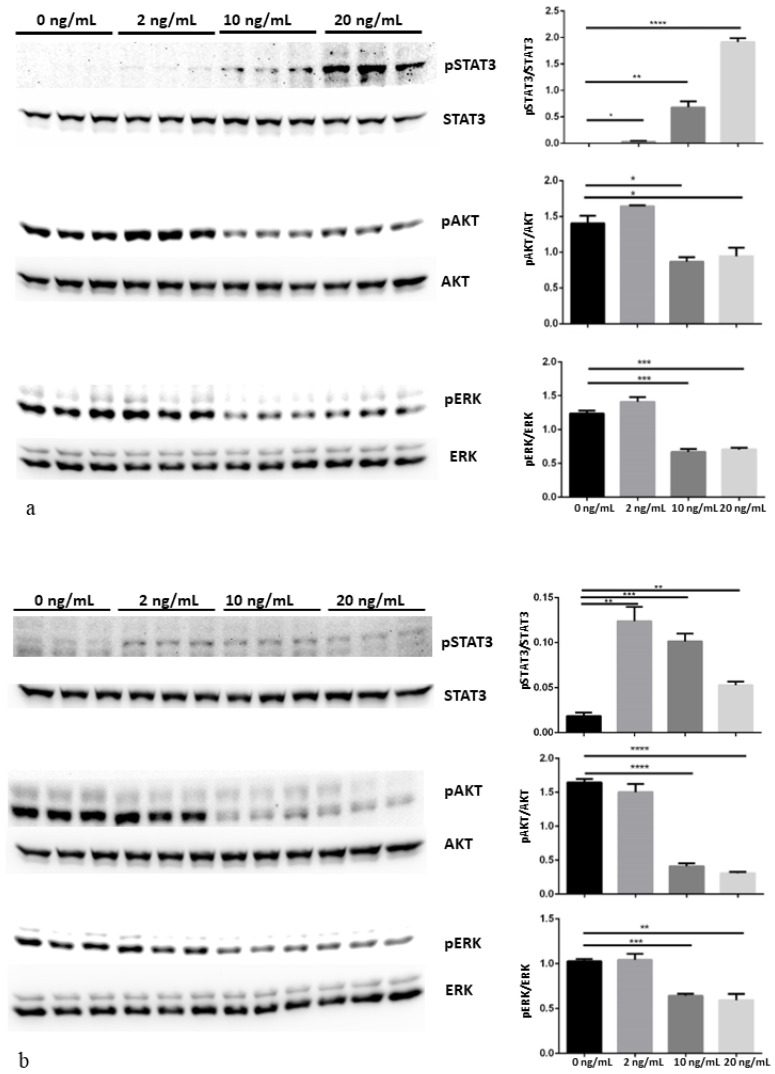
Dose-dependent response to rhCNTF of JAK/STAT3, PI3K/Akt and MAPK/ERK proteins pathways in LNCaP (**a**) and 22Rv1 (**b**) prostate cancer cell lines by Western blot analysis. LNCaP and 22Rv1 cell lines were treated with 0, 2, 10 and 20 ng/mL rhCNTF for 15 min. Bands were quantified and results were calculated in arbitrary units (AU) and reported as bars of a histogram. pSTAT3, pAKT and pERK1/2 quantities were normalized using total STAT3, AKT and ERK1/2, respectively. Data are represented as mean ± SD. * *p* < 0.05, ** *p* < 0.01, *** *p* < 0.001, **** *p* < 0.0001. Original images of immunoblotting data in the Appendix A.

**Figure 4 cancers-14-05917-f004:**
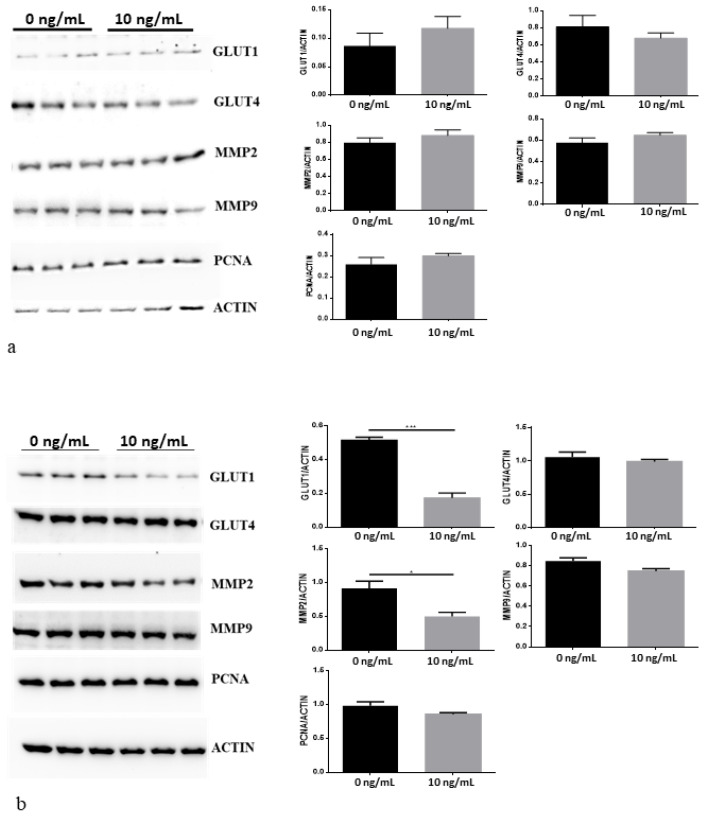
GLUT-1, GLUT-4, MMP-2, MMP-9 and PCNA expressions after treatment of LNCaP (**a**) and 22Rv1 (**b**) prostate cancer cell lines with 10 ng/mL rhCNTF for 24h. Bands were quantified and results were calculated in arbitrary units (AU) and reported as bars of a histogram. GLUT-1, GLUT-4, MMP-2, MMP-9 and PCNA expressions were normalized with β-actin expression. Data are represented as mean ± SD. * *p* < 0.05, *** *p* <0.001. Original images of immunoblotting data in the Appendix A.

**Figure 5 cancers-14-05917-f005:**
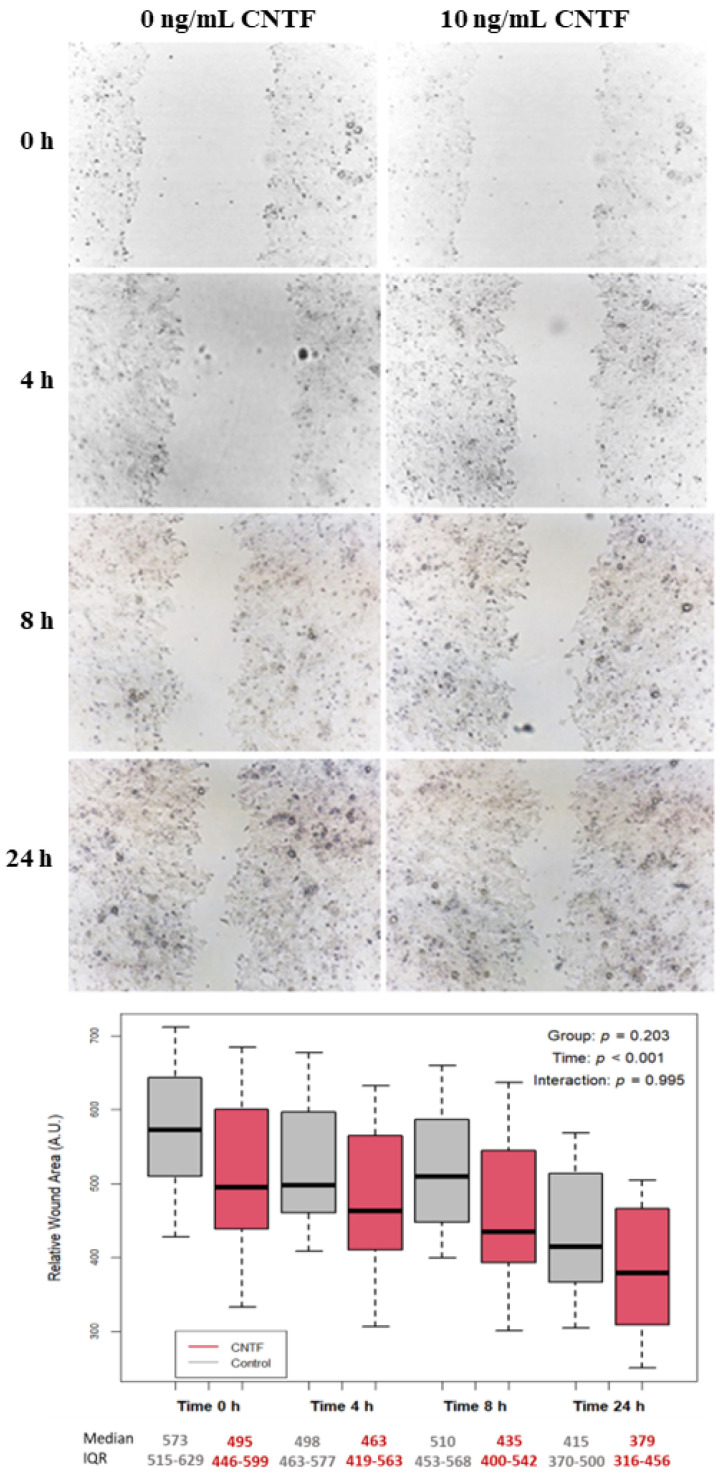
Wound healing assay after treatment with rhCNTF in 22Rv1 cell line. Representative photos of wound areas are shown; no significant difference in wound closure was found in cells treated with 10 ng/mL CNTF compared to untreated controls. Results are expressed in arbitrary units (A.U.) and reported as boxplots.

**Figure 6 cancers-14-05917-f006:**
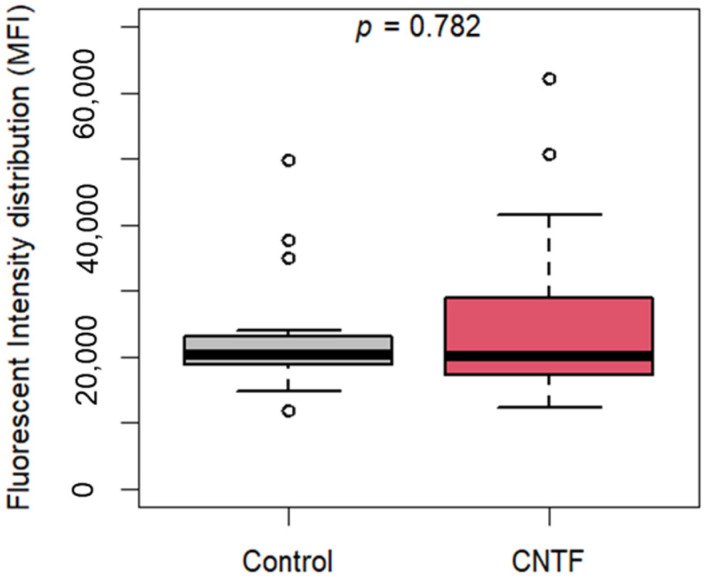
Glucose uptake after rhCNTF treatment of 22Rv1 cells. Boxplots show no differences in fluorescent intensity distribution (M.F.I.) between treated (median 20271, IQR 17416-28688) and untreated (median 20364, IQR 18784-23100) 22Rv1 cells.

**Figure 7 cancers-14-05917-f007:**
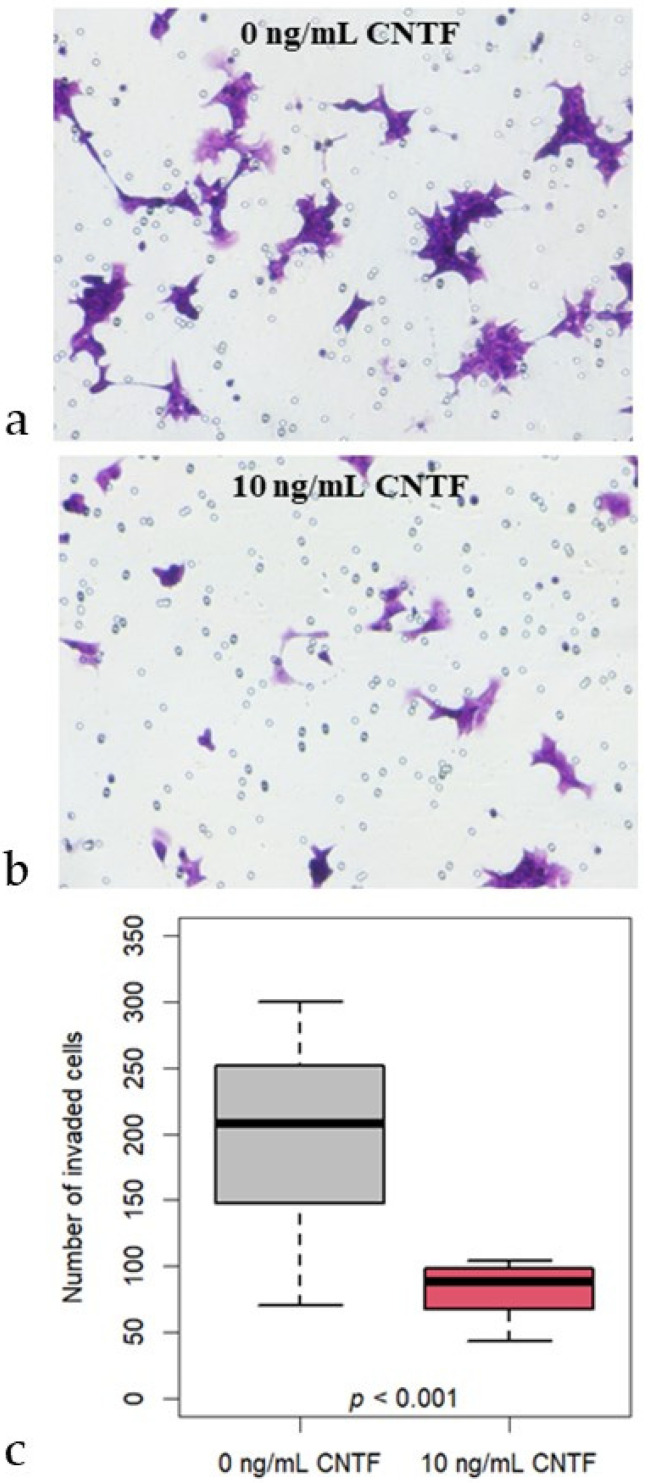
Transwell invasion assays. Representative images of 22Rv1 cells stained with crystal violet showing CNTF treatments effect: (**a**) untreated 22Rv1 cells; (**b**) 22Rv1 cells treated with 10 ng/mL rhCNTF; (**c**) Boxplot shows that the distribution of the number of 22Rv1 invaded cells is significantly (*p* < 0.001) lower in the group treated with 10 ng/mL CNTF (median 89; IQR 71-98) compared to the untreated controls (median 208; IQR 149-247).

**Figure 8 cancers-14-05917-f008:**
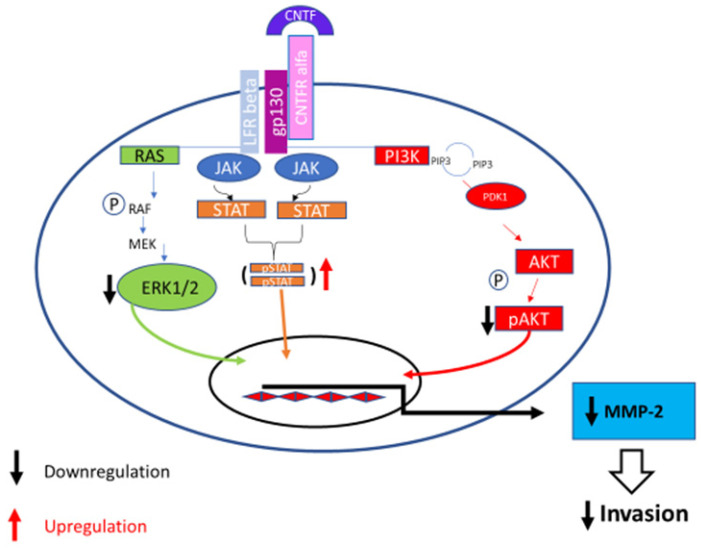
CNTF pathway in PCa. The binding of CNTF to the receptor actives JAK/STAT3 and downregulates MAPK/ERK, PI3K/AKT signaling pathways in CRPC cell model (22Rv1 cell line). Subsequently, MMP-2 is downregulated and cell invasion decreases.

**Table 1 cancers-14-05917-t001:** Antibodies used in this study.

Antibody	IHC	WB	IF	Company
pAb Rabbit anti-human CNTF (#ab190985)	1:500	//	1:100	Abcam, Cambridge, UK
pAb Rabbit anti-human CNTFRα (#PA5-45053)	//	1:400	//	Thermo Fisher Scientific, Waltham, MA, USA
mAb Mouse anti-human CNTFRα (#ab89333)	1:150	//	1:100	Abcam, Cambridge, UK
mAb Rabbit anti-human pAKT (#4060)	//	1:1000	//	Cell signaling technology, Danvers, MA, USA
pAb Rabbit anti-human AKT (#9272)	//	1:1000	//	Cell signaling technology, Danvers, MA, USA
mAb Rabbit anti-human pERK1/2 (#4377)	//	1:800	//	Cell signaling technology, Danvers, MA, USA
mAb Rabbit anti-human ERK1/2 (#4695)	//	1:1000	//	Cell signaling technology, Danvers, MA, USA
mAb Mouse anti-human pSTAT3 (#4113)	//	1:800	//	Cell signaling technology, Danvers, MA, USA
mAb Rabbit anti-human STAT3 (#4904)	//	1:1000	//	Cell signaling technology, Danvers, MA, USA
pAb Rabbit anti-human GLUT1 (#PA1-46152)	//	1:500	//	Thermo Fisher Scientific, Waltham, MA, USA
pAb Rabbit anti-human GLUT4 (#PA5-23052)	//	1:500	//	Thermo Fisher Scientific, Waltham, MA, USA
pAb Rabbit anti-human MMP-9 (#10375-2-AP)	//	1:500	//	Proteintech Group, Manchester, UK
mAb Mouse anti-human MMP-2 (#436000)	//	1:500	//	Thermo Fisher Scientific, Waltham, MA, USA
mAb Mouse anti-human PCNA (#sc-56)	//	1:250	//	Santa Cruz Biotechnology, Inc, Dallas, TX, USA

mAb: monoclonal antibody; pAb: polyclonal antibody; IHC: Immunohistochemistry; IF: immunofluorescence; WB: Western blotting.

## Data Availability

The data will be shared after the institutional approval.

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
