# Peer review of "Ciliary Neurotrophic Factor Modulates Multiple Downstream Signaling Pathways in Prostate Cancer Inhibiting Cell Invasiveness"

_cancers, 2022, doi:10.3390/cancers14235917_

Round 1

Reviewer 1 Report

Manuscript reviewed

Tossetta, G.; Fantone, S.; Gesuita, R.; Goteri, G.; Senzacqua, M.; Marcheggiani, F.; Tiano, L.; Mar-zioni, D.; Mazzucchelli, R. Ciliary Neurotrophic Factor Modulates Multiple Downstream Signaling Pathways in Prostate Cancer Inhibiting Cell Invasiveness. Cancers 2022, 14, x. https://doi.org/10.3390/xxxxx

Simple Summary

The sentence structures and use of grammar require attention: this section is not yet written to an acceptable standard for publication.  Unfortunately this section is particularly poorly written and requires major revisions.

Abstract

The sentence structures and use of grammar require attention: this section is not yet written to an acceptable standard for publication.The authors have done an excellent job summarizing their study and concisely highlighting their key findings in the field.  Once its written expression is brought up to standard, this Abstract is likely to evoke interest due to the knowledge gap it addresses being associated with very important signaling pathways.

Methods

The sentence structures and use of grammar require attention: this section is not yet written to an acceptable standard for publication.

Again, the authors need not address the content as this is a satisfactory summary of the study's methodology.

Introduction

The sentence structures and use of grammar require attention: this section is not yet written to an acceptable standard for publication.

That aside, the authors have provided a very astute introduction in terms of introducing the key signaling pathways, identifying relevant functions and knowledge gaps in a prostate cancer setting.  In doing so, their proposed studies are very well-justified.

Materials and Methods

Please address the following queries:

IHC and IF negative controls

With respect to negative controls for IHC in the PCa and CRPC prostate samples: why did the authors not used an isotype-matched primary antibody in favor of not omitting the primary or secondary antibodies?

With respect to negative controls for IF on LNCaP and 22Rv1 cells, again: why did the authors omit the use of an isotype-matched primary antibody control in favor of omitting the primary or secondary antibodies?

Having regard to both IHC and IF, about, what is the purpose of omitting the secondary antibody as a control on both occasions?

Determining optimal dose of CNTF

With respect to CNTF treatment of LNCaP and 22Rv1 cells in order to determine a dose/responsive curve, was this done in the presence or absence of serum?  Were the cells counted, seeded at a particular number of cells prepare tissue culture vessel, serum-starved before the treatment?  In which tissue culture vessels were the treatments done (e.g. 6-,12-, 24- well plates, T25 flasks, etc.)

With respect to CNTF treatment of LNCaP and 22Rv1 cells in order to determine a dose/responsive curve: was this done in the presence or absence of serum?  Were the cells counted, seeded at a particular number of cells per tissue culture vessel?  Were the cells serum-starved before the treatment?  In which tissue culture vessels were the treatments done (e.g. 6-,12-, 24- well plates, T25 flasks, etc.)

Details of wound healing assay

With respect to the wound healing assay, what was the status of serum in the medium after the cells were washed and treated with rhCNTF?

Statistical analysis

I am surprised that the data in the case of all assays were not normally distributed such as to warrant the use of non-parametric testing.  I can understand this would be the case if the authors were comparing intensity across IHC and IF data, especially when comparing data obtained from the prostate cancer patients' biopsies and analyzing intensities between PCa and CRPC tissue samples.

However, almost all studies I am aware of that utilized the scratch assays assume Gaussian distribution and run independent t-tests and one way ANOVA.  There are no reasons that I am aware of as to why this assay should be analyzed in any other way.

As a completely random example of a study from which I am entirely independent:

Zheng A, Bilbao M, Sookram J, Linden KM, Morgan AB, Ostrovsky O. Epigenetic drugs induce the potency of classic chemotherapy, suppress post-treatment re-growth of breast cancer, but preserve the wound healing ability of stem cells. Cancer Biol Ther. 2022 Dec 31;23(1):254-264. PMID: 35389825; PMCID: PMC8993057.  The authors state:

"Statistical analysis

All experiments were repeated three to seven times. The number (n) of repeated experiments is listed in the figure legends. Statistical analyses of viability were taken with independent t-tests and one-way analysis of variance (ANOVA) testing. P < .05 was used for statistical significance. Data analysis was performed using SPSS Statistics software version 22 (IBM, Armonk, NY)." (See Figure 5.)

I implore the authors to analyze their data arising from their scratch assays consistent with the overwhelming amount of studies that utilize statistical tests assuming normal distribution.

Results

The authors have performed each of the studies to a very high standard, the data in each of the figures are very clear and well thought out.

I do not have a problem with any of the data as it is presented in its current form, save for the statistical tests used to analyze the scratch assays.

Discussion

The discussion has elements of poor structure, some sentences require redrafting and there are quite a few grammatical errors.  It also needs to be broken up into paragraphs, instead of one whole block of text, which is likely to lose the reader.

Otherwise, the content of the discussion and interpretation of the data is good and the authors do clearly state when they are being speculative.  However, the authors need to to be very careful not to overstate their findings and to identify the study's limitations.

For example, the authors saw a correlation between the inhibition of MMP-2 and a reduction of cell invasion in the transwell assay.  However, this is far from definitive.  The authors do not over express MMP-2 in the 22Rv1 cell line to see whether the reduced invasion can be rescued by restoring MMP-2 expression in the CNTF treated cells. Neither do the authors run zymograms to confirm a reduction in MMP-2 activity towards its collagen/gelatin substrate.

The authors are required to discuss this limitation to make it clear to the reader that whilst the data strongly suggest MMP-2 to be the main driver of invasion, the data they currently have is equivocal and further experiments are required but were outside the scope of the current 'proof of concept' study.

Conclusion

Lastly, the authors have completely overstated the implication of their findings to the clinical setting.  This study has a very long way to go before statements as bold as that in the conclusion can safely be made.  More studies are clearly required, such as in vivo male mouse studies in which 22Rv1 (and control) cells are to be injected and tumor growth and metastasis is analyzed in groups of mice +/- daily administration of CNTF. 

In short, it is not appropriate to jump straight from in vitro data to the importance of CNTF in therapies against androgen insensitive prostate cancer in humans.

Overall comment

The authors have presented a scientifically sound study pertaining to an important cell signaling pathway in which they have begun to address a clear knowledge gap.  With further studies to confirm the mechanism(s) underlying reduced prostate cancer cell invasion and studies of androgen insensitive prostate tumor growth and metastatic dynamics in mice treated +/- CNTF this proof-of-concept study could lead to novel therapeutic intervention in prostate cancer patients.

However, the standard of the written presentation of the study in its current form significantly detracts the relative importance of the study in the relevant field.  Major improvements are required before this study can be considered to meet the standard for the purposes of publication.

***

Reviewer 2 Report

I am a HemOnc who has focused on PCa for 40 years. I found this info on CNTF to be of great interest-- and I enjoyed reading the authors' manuscript.  I feel this should be published.  There are significant issues with English grammar involving sentence structure, etc.  I converted the PDF sent to me into a Word document and used the Reviewer tool in word to show all areas I offered corrections to enhance the readabilty of this work.  

My major issue with the many publications on similar findings is why is there so great a problem getting such findings translated into patient evaluation and management? Everyday of my professional life I come across exciting developments that possibly would dramatically alter lives of people-- and yet this is rarely done.  CNTF has huge ramifications in many areas of medicine, and because of the authors' work I pursued many of the citations and went beyond those and downloaded many papers that further encourage an intense focus on ciliary neurotrophic factor (CNTF). 

I will submit to MDPI the Word doc showing markup changes, the same document with markup accepted and a PDF of my changes as well.  Please excuse the occasional page that appears blank due to formatting issues. 

Round 2

Reviewer 1 Report

The authors have sufficiently addressed all concerns.  The manuscript is significantly improved and is now at the standard required for publication.

Author Response

Thank you to reviewet for his comments.

Reviewer 2 Report

I would love to see a graphic portrayal of how the authors see the CNTF pathway in the context of PCa, showing areas of inhibition, up- and down-regulation.  I have searched for such an image and cannot find one in this context.  A pathway image is shown, but it does not exhibit the inhibition as proposed by the authors. https://www.rndsystems.com/pathways/cntf-signaling-pathways#print-copy-form

Author Response

Thank you to reviewer for the suggestion.

We added the Figure 8 concerning CNTF pathway in PCa before the conclusion paragraph.
